# The Importance and Scientific Value of Long Weather and Climate Records; Examples of Historical Marine Data Efforts across the Globe

**Jürg Luterbacher** [1,*], **Rob Allan** [2], **Clive Wilkinson** [3], **Ed Hawkins** [4], **Praveen Teleti** [4], **Andrew Lorrey** [5], **Stefan Brönnimann** [6], **Peer Hechler** [1], **Kondylia Velikou** [7] **and Elena Xoplaki** [8]

1    World Meteorological Organization (WMO), 1211 Geneva, Switzerland; phechler@wmo.int
2    Institute for Environmental Research & Sustainable Development (IERSD), National Observatory of Athens, 11810 Athens, Greece; allarob@gmail.com
3    Department of Meteorology, University of Reading, Reading RG6 6UR, UK; clive.wilkinson@reading.ac.uk
4    National Centre for Atmospheric Science, Department of Meteorology, University of Reading, Reading RG6 6UR, UK; ed.hawkins@ncas.ac.uk (E.H.); p.r.teleti@reading.ac.uk (P.T.)
5    National Institute of Water and Atmospheric Research, Auckland 1149, New Zealand; andrew.lorrey@niwa.co.nz
6    Institute of Geography and Oeschger Centre for Climate Change Research, University of Bern, 3012 Bern, Switzerland; stefan.broennimann@unibe.ch
7    Department of Meteorology and Climatology, School of Geology, Faculty of Sciences, Aristotle University of Thessaloniki, 54124 Thessaloniki, Greece; kvelikou@geo.auth.gr
8    Department of Geography and Center for International Development and Environmental Research, Justus Liebig University of Giessen, 35390 Giessen, Germany; elena.xoplaki@geogr.uni-giessen.de
*    Correspondence: jluterbacher@wmo.int

**Abstract:** The rescue, digitization, quality control, preservation, and utilization of long and high quality meteorological and climate records, particularly related to historical marine data, are crucial for advancing our understanding of the Earth's climate system. In combination with land and air measurements, historical marine records serve as foundational pillars in linking present and past weather and climate information, offering essential insights into natural climate variability, extreme events in marine areas, baseline data for assessing current changes, and inputs for enhancing predictive climate models and reanalyses. This paper provides an overview of rescue activities covering marine weather data over the past centuries and presents and highlights several ongoing projects across the world and how the data are used in an integrative and international framework. Current and future continuous efforts in data rescue, digitization, quality control, and the development of temporally high-resolution meteorological and climatological observations from oceans, will greatly help to further complete our understanding and knowledge of the Earth's climate system, including extremes, as well as improve the quality of reanalysis.

**Keywords:** historical marine data; data rescue and digitization efforts; early meteorological and climatological observations; Earth's climate system; reanalysis

## 1. Introduction

Weather data have been collected on an unsystematic basis since the late 17th century, often attached to weather and climate events with considerable impact on economies and societies. The history of coordinated weather observations by an observational network-dates back to more than 200 years ago when, in 1781, the Societas Meteorologica Palatina in Europe began systematic and coordinated weather observations [1,2]. Many measurements have been taken since then, but few meteorological observing stations have been operated from the same place over decades or centuries without disruption, e.g., ref. [1] and references therein. Such long-term observing stations represent a real heritage, and their time series of observational data represent unique sources of knowledge. There is

no other source of systematic historic data for analyzing and understanding the status, physical characteristics, and spatiotemporal variability of the atmospheric elements of the climate system [3]. The value of old historical weather observations in understanding the Earth's climate system cannot be overstated. These records, often dating back centuries, offer a unique and invaluable perspective on past weather patterns, climatic variability, and extreme events. The significance of these historical weather observations lies in their ability to complement modern climate data, providing crucial insights into long-term climate trends, variability, and the drivers behind climatic shifts. One recent example of an end-to-end approach to demonstrate the value of specific old meteorological measurements is shown by [4], who discuss some ship pressure observations in helping to determine the strength of a cyclone that hit Western Australia in 1921, where existing measurements are sparse to non-existent.

Long-term observations from meteorological and climatological stations are vital inputs to reanalysis [5–7] as well as climate models. Historical reanalyses, like the global Twentieth Century Reanalysis (20CR, [6,7]), which assimilate terrestrial and marine air pressure observations, apply strong initial quality control assessments to these data prior to and during their assimilation. 20CR generates 80 equally likely members that capture the uncertainty due to the initial conditions as well as the sea–surface temperatures used as boundary conditions given the assimilation system. This allows for investigations of the initial input data to be assimilated into the reanalysis, in order to isolate and correct any data problems, or deem the data to be too unreliable for use.

Historical weather observations also serve as a fundamental component in reconstructing past meteorological features and placing them in the context of current conditions. These records, documented in various forms such as handwritten journals, logbooks, diaries, and early instrumental measurements, offer insights into weather conditions predating the establishment of standardized meteorological networks. They contribute essential information about temperature, precipitation, wind (speed, direction), pressure, marine conditions, and specific meteorological phenomena.

Data rescue, and specifically marine data rescue, is a process by which data from original historic documents are converted to a machine-readable format. The process is many faceted, beginning with the finding and assessment of the original documents in the archives. This is followed by scanning or photography, after which the data are keyed, either directly or through crowdsourcing, or by Artificial Intelligence and optical character recognition in the future (once the necessary technology is perfected). The resulting output is then formatted, processed, and quality controlled, before being made available to the scientific community. There are good practice guidelines available through the WMO [8] and the Copernicus Climate Services portal (https://climate.copernicus.eu/sites/default/files/2020-02/BestPracticeGuidelines_ClimateDataRescue_0.pdf (accessed on 8 January 2024)) for both imaging and keying original documents, and some key points are listed below. A joint recent effort aims at merging C3S and WMO guidelines on data rescue.

The main recommendations are as follows:

1. All original documents should be imaged in their entirety.
2. Images of the original documents should be securely preserved and made easily accessible so that the provenance of every observation can be verified.
3. It should be clearly documented as to whether an original document has been imaged, keyed, and processed as this will avoid needless duplication of effort in the future.
4. Ideally, all observations should be keyed from a document.
5. Where it is not possible to key all observations, due for instance to time or financial constraints, then this should be well documented and made clear.
6. All instruments and observing metadata should be keyed as well as metadata concerning the observing platform.

Marine data present a number of challenges connected with instrument heights and exposure, changes in the size and design of vessels (the observing platform), changes in instrument types and in observing methodology. Some of these challenges are well

documented in the literature, especially for instance measurements of SST. The SST data can be influenced by the type of bucket in use, the use of thermometers in engine room intakes (ERIs), and, more recently, hull sensors.

More work is needed on potential biases arising from the exposure of barometers and thermometers on board ships, but this is entirely dependent on the collection and analysis of metadata that record the instruments, their placement, and the observing methodology. Such work is underway and requires the examination and documentation of almost every page of every ships' logbook. In some logbooks, there is extensive information on the instruments, but in many there is little if any metadata available, or the description of the instrument exposures are sketchy or vague. However, this is where the careful documentation of all metadata, however vague or apparently useless, is essential. In a series of logbooks for the same vessel, the collation of vague but varyingly described information can arrive at a more useful picture of instrument exposure. Additionally, a series of consecutive logbooks for the same vessel need to be examined together as it is often the case that the first logbook in a series will have a good description of the instruments, whereas the subsequent logbooks will contain little or no instrument descriptions. If you only image and key the subsequent logbooks without the first logbook, the metadata can be lost.

Information on thermometer screen exposures can in some cases be gleaned by examining the relative positions of other instruments. For instance, in the late 19th century many merchant and naval vessels made use of deck houses or other forms of superstructure. A chart house would frequently be where the barometer was placed and its height above the sea was always given. The thermometer screen might then be placed on the outside of the chart house, usually four feet above the deck, and therefore within a foot or two of the height of the barometer.

Similar information can be found by a careful reading of the logbook itself. Thermometer screens, if well exposed, were always in danger of being washed away in severe weather, and where such instances are recorded in the log, there are usually descriptions of the placement of the original screen and what steps were taken to find a more secure location. Other screen exposures were clearly unsuitable, for instance at the head of a companion, or at the break of the poop where, on sailing vessels in particular, they were too well sheltered and therefore subject to potential ship heating. The reason for them being so placed was to avoid having them washed away in heavy seas. It is important to know where these screens were placed, even if their location was not ideal.

Further, a close examination of the logbook can reveal how sea temperatures were recorded, although these accounts are extremely rare. Sometimes the type of bucket is mentioned. There are accounts of the seawater being pumped into a bucket rather than being drawn in a bucket directly from the sea; however, in such instances, the water was pumped for many minutes before a sample was collected. This may have been widely practiced until instructions were issued by the British Board of Trade not to do so. This directive, and others of a similar nature, were printed in the observing instructions found at the front of each logbook. Examining the printed instructions and observing any amendments and additions over a period of time can also inform us as to observing methods and changes in those methods. The points above and many others not mentioned here are the starting point for considering any adjustments to or potential bias issues in the observations. The essential first step is to collect, collate, and record all instrument metadata, no matter how incomplete or vague. Collecting this information is as important as the observations themselves.

Although much valuable data rescue has been performed in the past, this has often been a component and part of the output of a research project. Such projects are usually narrowly defined, meaning that the data rescue component is constrained by time and financial boundaries, as well as the data needs of the project itself. It is therefore essential that marine and other data rescue efforts are treated as projects in themselves or even as a program, where the sole focus is the gathering of all observations recorded in collections. In

addition, a good mixture of sail and steam vessels is needed to get comprehensive spatial coverage of observations across the world's oceans.

Over the past two decades, marine data rescue has also adopted a strategy that has addressed, as far as possible, a need to infill major data gaps in the historical record. This has included the seas in both the Arctic and Antarctic regions, as the least well represented in global data sets. There has also been a major focus on the Pacific, and to a lesser extent the Indian Ocean. This addressed an urgent need to gather data for the sparsely represented Southern Hemisphere.

In order to make the most of limited financial resources, it was clear that sets of logbooks that undertook long voyages, for instance to the Pacific, or voyages of circumnavigation, provided the most cost-effective way of gathering large amounts of data over a wide area of the globe. This was the most efficient way of obtaining more data for the Southern Hemisphere but had the added bonus of contributing more data to the Northern Hemisphere record as well. For instance, the various passages on a voyage from Finland to Hawaii provided data for the Baltic, North and South Atlantic, and the South and North Pacific. Similarly, a voyage from Britain to Japan provided passages through the North and South Atlantic, the Indian Ocean, and parts of the North Pacific. This mode of working also ensured that, unlike in the past, data from an entire voyage was captured instead of data only being keyed for a geographically specific area. In a similar vein, Ref. [9] presented a comprehensive new compilation of cyclone activity in North America and the Caribbean for the second half of the 19th century using more than 9000 newspaper marine shipping news reports and other unique land and marine data in order generate unique weather maps used in historical tropical cyclone research.

By rescuing, digitizing, and analyzing these records, researchers can extend climate datasets far beyond the era of modern instrumental observations [1]. This extension facilitates a more comprehensive understanding of natural climate variability and trends. These records help also identify recurrent patterns, spatial variations, and regional climate sensitivities, enabling better preparation and adaptation to future extreme weather risks. The interdisciplinary nature of historical weather observations further amplifies their significance. These records often include qualitative descriptions of weather phenomena, ecological observations, agricultural records, and societal impacts of climatic changes. Such information helps with understanding climate dynamics as well, linking the interactions between climate, ecosystems, and human societies across different time and space scales. Further, historical weather observations are indispensable for assessing the frequency, intensity, and duration of extreme weather events. Studying past storms, droughts, heatwaves, and cold events provides crucial context for evaluating the changing behavior of extreme events in a warming world.

Furthermore, historical weather observations play a crucial role in validating and refining climate models. By comparing model simulations with past weather patterns obtained from historical records, scientists can assess the models' accuracy in capturing known climatic variations. This validation process strengthens confidence in future climate projections and helps identify areas where models require refinement. A recent compilation of early instrumental data [10], which additionally contains 13,822 station years of newly digitized data, is now available for climate reconstructions. Complemented with proxy and documentary data, they allow new global data products based on data assimilation [11] several centuries back.

## 2. Supportive International Activities

Long-term, high quality, and reliable instrumental climate records are indispensable pieces of information required for undertaking robust and consistent studies to better understand, detect, predict, and respond to global climate variability and change [12]. As one example, Figure 1 shows the annual mean temperature at Hohenpeissenberg, Germany, covering the period 1781 to 2022. Maintaining the operation of historically uninterrupted

stations and observing systems has been acknowledged as one of the key principles of climate monitoring [13,14].

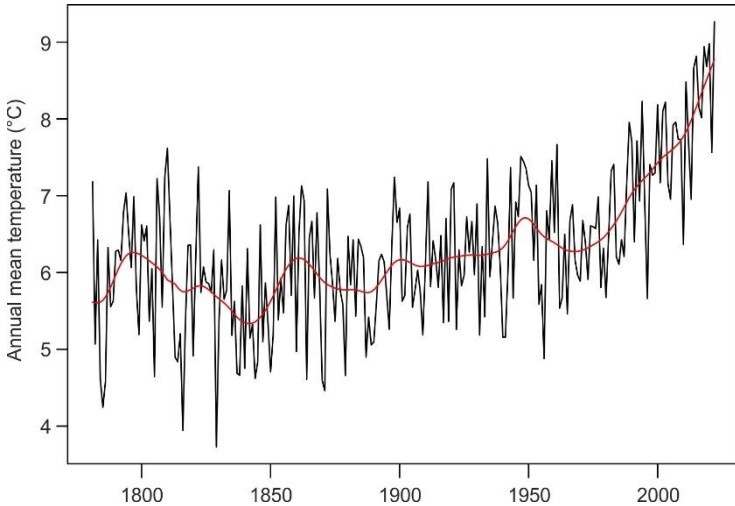

**Figure 1.** Annual mean temperature at Hohenpeissenberg from 1781 to 2022 (red: cubic smoothing spline with a smoothing parameter of 0.57, corresponding approximately to a 30-year average). The temperature increases from 1781 to 2022 amounts to 3.16 °C. source Deutscher Wetterdienst, DWD.

In 2013, the WMO Executive Council urged Members to sustain observation programs in support of centennial observations (Figure 2, exemplified with the Sonnblick Observatory, Austria) as an invaluable scientific heritage for future generations. The Council requested WMO Technical Commissions to investigate existing site certification mechanisms, network criteria, and monitoring principles and to set up an appropriate WMO mechanism for the recognition of centennial observing stations, based on a minimum set of objective assessment criteria [15].

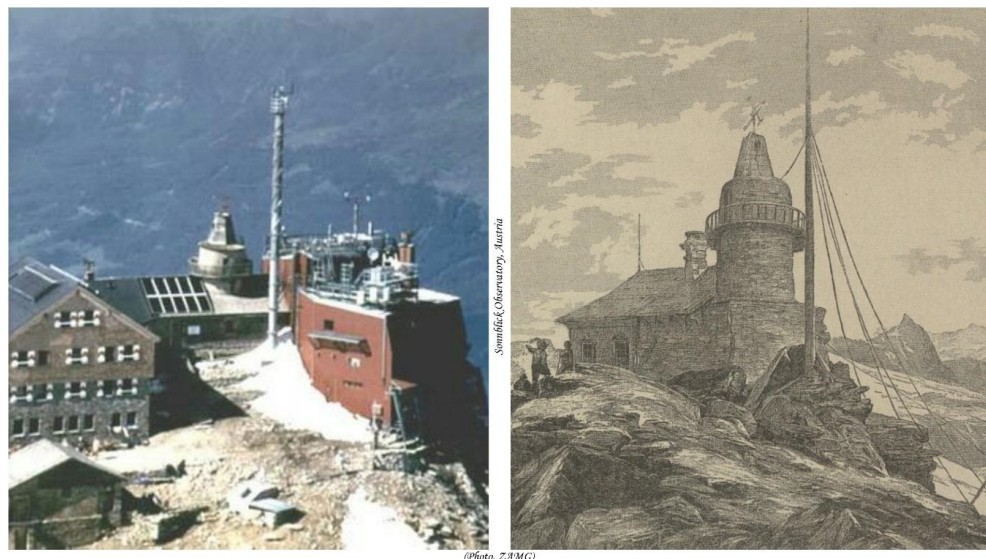

**Figure 2.** Sonnblick Observatory, Austria, 2001 (**left**) and 1886 (**right**), source GeoSphere Austria (former ZAMG).

Based on the outcomes of a WMO scoping meeting on a potential WMO recognition mechanism for centennial observing stations in June 2014, the 17th World Meteorological Organization Congress decided to develop a recognition mechanism for long-term observing stations, including centennial observing stations, and the possibility of intermediate-level certification for 50 years and 75 years of observations [16]. Following the successful conduct

of a test phase, showing that 34 Members representing all six WMO regional associations had responded and submitted 79 candidate stations, the WMO Executive Council decided to endorse the mechanism and criteria for the WMO's recognition of (meteorological) long-term observing stations [17]. The first set of 60 centennial observing stations had been endorsed by the WMO Executive Council in 2017 [18], followed by a second set of 57 centennial observing stations in 2018 [19]. A dedicated WMO website related to centennial observing stations was implemented and has been updated regularly since then (https://wmo.int/centennial-observing-stations (accessed on 8 January 2024)). In 2019, WMO experts held a meeting to further develop the WMO recognition mechanism by analyzing the experiences made so far. Consequently, the initial WMO recognition mechanism and its criteria had been refined in 2020 and 2021 [20,21], and the mechanism broadened in 2023 to include centennial marine and hydrological observing stations and a possibility to nationally recognize 75+ years stations [22]. In parallel, another 293 centennial observing stations have been recognized by the World Meteorological Congress and WMO Executive Council [20,21,23], and the first edition of a series of State of Recognition reports were published in 2022 [3]. All in all, 406 centennial observing stations have been recognized by summer 2023 (10 centennial marine observing stations, 22 centennial hydrological observing stations, and 372 centennial meteorological observing stations).

Long-term observations greatly contribute to WMO flagship products, such as the annual global and regional State of the Climate reports, which provide scientifically sound, reliable information for policymakers and decision makers. WMO has produced the annual State of the Global Climate report since 1993 (https://wmo.int/publication-series/state-of-global-climate accessed on 8 January 2024), which is now complemented by regional reports. Global estimates and analyses require both in situ data and historical observations provided by WMO Members. Among these records, historical marine data represent a treasure trove of information that has the potential to significantly enhance our understanding of climate dynamics.

Historical marine data encompass a rich array of information gathered from ships' logs, scientific expeditions, and marine observations dating back centuries. These data contain invaluable observations of sea–surface temperatures, weather patterns, ocean currents, ice cover, biological phenomena, and more. The spatial sampling of ship data makes their quality assurance more difficult than for land data. Many factors need to be considered, ranging from ship height to the placing and exposition of the instruments, interpolation of ship coordinates, and possible errors in correctly understanding ship logs. However, for marine air temperature data, approaches have been developed to assess biases and errors and incorporate them into error models and uncertainty estimates (e.g., [24]). Marine air pressure measurements that are used as input into reanalysis may have to be debiased offline or during the assimilation procedure [6]. In addition, much of this historical marine data remains scattered across archives, libraries, and repositories worldwide, often in fragile or deteriorating formats. The rescue and digitization of these invaluable records represent an urgent priority for climate researchers. By rescuing, digitizing, and standardizing these historical marine datasets, we can unlock a treasure trove of information, enabling scientists to extend climate records further back in time and expand spatial coverage. It should also be noted that the entirety of a logbook is imaged and keyed; this also includes all passages a vessel made in the course of a voyage. Thus, these efforts hold immense promise in refining our understanding of past climate conditions globally and improving the accuracy of climate models used for future projections.

This contribution builds and expands on the publication of [25] and others, highlighting some examples of new data sources, regional data activities, and the need for good metadata, high standards, and quality control of historical marine weather observations covering the past centuries. Much of this has been made possible by the international ACRE (Atmospheric Circulation Reconstructions over the Earth, www.met-acre.net accessed on 8 January 2024) initiative and its specific ACRE Oceans chapter, with strong links to the International Comprehensive Ocean-Atmosphere Data Set (ICOADS), the Global Surface

Air Temperature (GloSAT) (https://www.glosat.org/ accessed on 8 January 2024) projects and Copernicus Climate Change Service (C3S) and the associated UK funding.

## 3. Examples of Historical Marine Data Capturing Efforts across the Globe Covering the Past Centuries

ICOADS is the prime data source for observations of marine air temperatures, sea–surface temperatures, sea level pressure, and several other "essential climate variables (ECVs)". The data coverage spans the globe and extends back in time to the late eighteenth century, albeit with increasingly sparse data coverage further back in time. ICOADS is the main source of observations for the marine component of atmospheric reanalysis. Its importance and utility cannot be overstated.

Although ICOADS holds a vast number of marine observations, these are not the entirety of what is available. There are many sets of original documents, such as ship logbooks, hydrographic reports, and a host of related material that is yet to be digitized and aggregated. These documents are to be found in naval and maritime libraries and museums, national archives, research institutes, and the archives of national weather services around the globe. Many such collections have been documented and cataloged but presently remain undigitized and thereby unavailable to the scientific community. Billions of observations are currently lost to science and need to be prioritized for their utility for different applications (see Table 5 in [26]).

Furthermore, many of the collections and source datasets (termed as decks as many were derived from data stored on punch cards) that are already incorporated into ICOADS are incomplete or fragmented. This is due to the gradual accretion of observations in ICOADS over past decades, from a diverse range of marine datasets, produced by other agencies. These datasets may have been produced by researchers or agencies to answer specific scientific questions, rather than to gather a broad range of data. Thus, these datasets tend to be spatially specific, or only contain certain subsets of observations, whereas the source documents may be much broader in scope.

## 4. Recent Advances in Historical Marine Data Rescue

A prime example of this is ICOADS Deck 201, the UK Marine Data Bank 1850–1920. The observations in this deck are based on the collection of ships' meteorological logbooks held by the UK National Meteorological Archive in Exeter. There are approximately 15,000 sets of logbooks covering this period and although not all of these logbooks are in Deck 201, many of them are. However, none of the more than 10 million sub-daily pressure measurements in these logbooks have been keyed and are therefore not in ICOADS. Furthermore, air and sea temperatures have only been selectively keyed, conforming to spatially specific areas (see Figure 3 showing data coverage from ICOADS deck 201 during the 1870s). The plot shows that large parts of the North Atlantic and Indian Ocean are devoid of observations and the South Atlantic has been omitted throughout the 1850–1920 period. Other parameters such as winds are similarly compromised, and observations of specific L gravity and ice have not been keyed. Other ICOADS decks have not yet been subject to similar scrutiny or comparison with their original documents, but it is likely that other decks are also compromised in a similar way.

In addition, there is frequently an absence of good instruments and observing metadata. Sometimes the metadata are missing from the original documents but often metadata are lost using compact data formats or because it was not thought to be important. The inadequacies arise from the original documents themselves, or more often, the project that produced a particular ICOADS deck, and in no way reflect on the achievement of ICOADS itself. There are valuable lessons to be learned here. Current and future marine data rescue must ensure that some of the issues raised above are addressed.

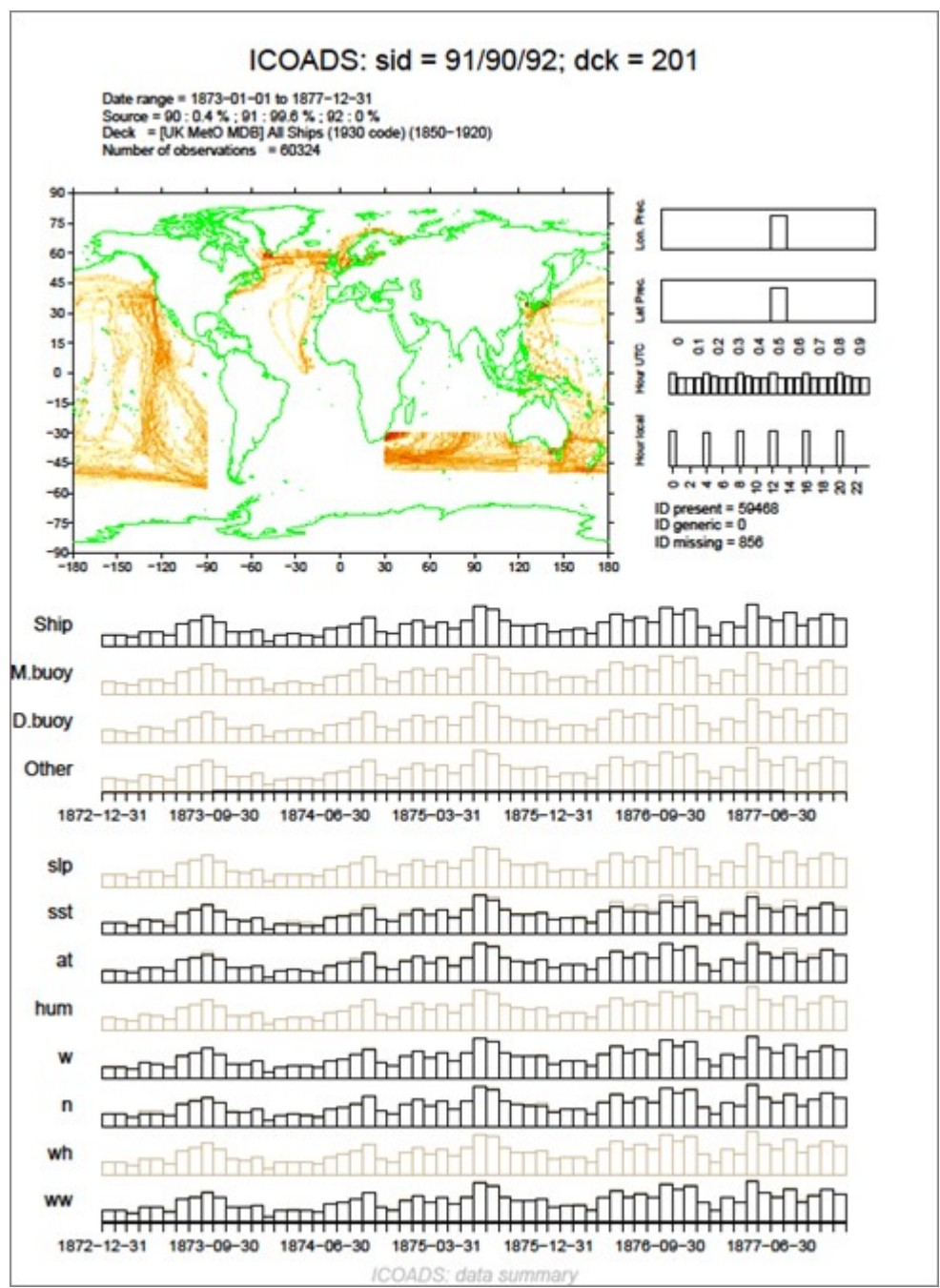

**Figure 3.** Global data coverage from ICOADS Deck 201, UK Marine Data Bank Spatial Coverage of Observations 1873–1877.

## 5. Some Specific Marine Data Sets and Regional Foci; Convict and Settler Ships Sailing to Australia and New Zealand

### 5.1. Convict Ships

From 1788 to 1868, some 806 ships sailed from England to Australia transporting male and female convicts. On board such vessels were Royal Navy surgeons and assistant surgeons who were required to compile and submit, at the end of each voyage, their journals and diaries detailing the medical health, treatment, and survival of the convicts during their journeys.

The UK National Archives (TNA) holds the above material in their ADM-101 series of over 1000 Royal Navy Medical Officer journals. In June 2008, the TNA was successful in a bid to the UK Wellcome Trust's Research Resources in Medical History Program to

fund the cataloging and scanning of these marine medical journals. The scans that have been made widely available through the online UK Ancestry WWW pages, cover the periods 1817 to 1856: https://www.ancestry.co.uk/search/collections/2318/ accessed on 8 January 2024 and 1858 to 1868: https://www.ancestry.co.uk/search/collections/2320/ (accessed on 8 January 2024). Those made on ships carrying female convicts are being transcribed/digitized by the volunteers of the Female Convicts Research Centre Inc. (https://www.femaleconvicts.org.au/convict-ships/convict-ship-records) (accessed on 8 January 2024) based in Tasmania.

Some of the above surgeons also made, and documented in considerable detail, various meteorological observations during their passages to the Antipodes. Depending on the instruments they possessed themselves and their interests, these observations ranged from a few annotations of observed air temperatures and pressures in page margins, or between descriptions of medical activities (see example in Figure 4), through to pages of tabulations of coarse monthly averages to detailed sub daily measurements of their ship's latitude, longitude, course, wind direction and magnitude, internal hospital and/or deck air temperatures, and barometric pressure. The latter that have survived, and passed quality control tests, are proving to be often the only records of instrumental weather at the time and at locations in the South Atlantic, southern Indian, and southern Australian waters.

**Figure 4.** Sample of a surgeon's journal with Barometer 30″ Temperature 82F daily meteorological observations amongst the text in light red.

At the end of these voyages, landfall in Australia was made either at Port Jackson near Sydney in New South Wales or Hobart in Tasmania. In the latter years of transportation following the Crimean War (1853 to 1856), most voyages ended at the port of Perth at Fremantle in Western Australia.

In the above online scans, the international ACRE initiative was able to isolate instances of daily to sub-daily instrumental meteorological observations made by ship surgeons on 158 convict ship voyages between 1817 and 1868. Of the observations that were found and digitized, 90 ships made only air temperature observations and 53 ships made both air temperature and barometric pressure observations. The logs with the remaining observations, though said to have been scanned, have so far not been found.

An example of the once-daily air temperature and barometric pressure observations made by a ship's surgeon along the track of a convict vessel sailing to Australia from England is shown in Figure 5, in this case the ship *Albion* in 1828. Instances where strong dips in the barometric pressure were observed could be matched with written accounts of severe weather and storms in the surgeons' journal, proving a basic initial check on the validity of the pressure record. On this occasion, no mismatches of observed barometric pressure falls and entries of severe storms were observed.

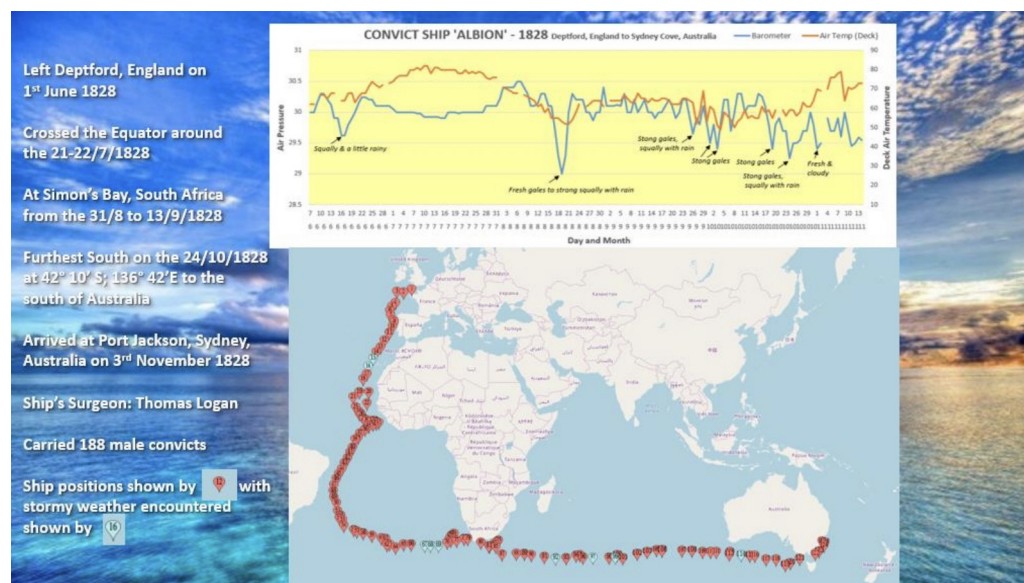

**Figure 5.** Map showing the route of the convict ship *Albion* in 1828 from England to Australia, with the once-daily air temperature (in red) and barometric pressure (in blue) observations plotted in the graph above. Strong dips in the barometric pressure in the observed pressure record matched accounts of severe weather and storms in the logs.

### 5.2. Settler Ships

In the British establishment of its colonies in Australia and New Zealand during the first half of the 19th century, what might be termed settler ships transported British migrants, who occasionally had some scientific, natural history, or official administrative background and made daily to sub-daily meteorological observations using thermometers and barometers. Two examples are shown below for *HMS Buffalo*, which transported the first European settlers to the state of South Australia in 1836 (Figure 6), and the ship *Tory*, which carried early European settlers to New Zealand in 1839 (Figure 7).

As with convict ships, the weather observations from settler ships often provide the only records of instrumental weather at the time and at locations in the South Atlantic, southern Indian, and southern Australia waters. By the mid to late 19th century, as the first cruise ships began to make journals between Europe and the Antipodes, weather observations, usually extracted from the ship's logs during voyages, began to appear in some of the onboard ship newspapers. These also begin to complement the weather observations made on other, often regular, voyages of mail or packet ships, cable laying ships, and yachts traveling around the world.

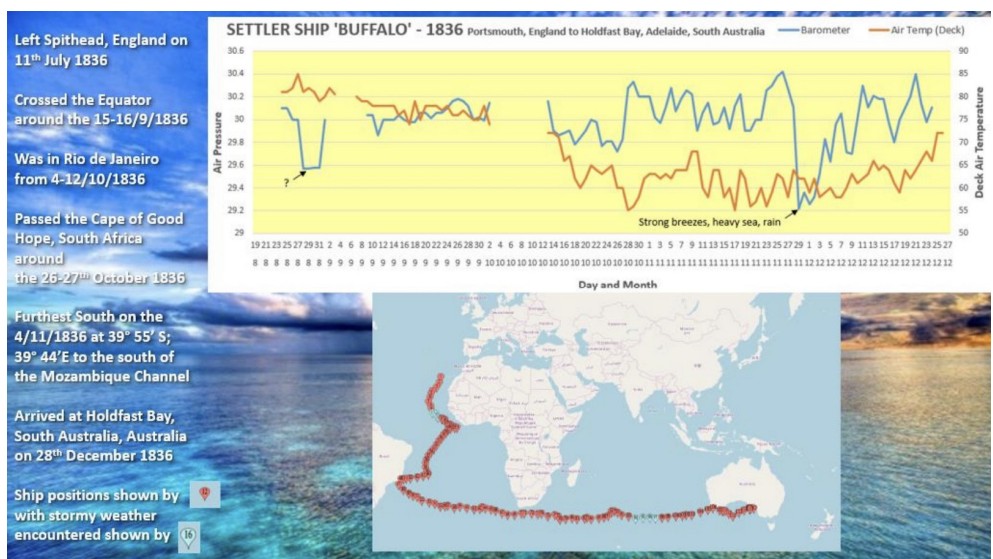

**Figure 6.** Map showing the route of the settler ship *HMS Buffalo* in 1836 from England to South Australia, with the once-daily air temperature (in red) and barometric pressure (in blue) observations from an abstract of the ship's logbook plotted in the graph above. The abstract log observations were made by Charles Brown Fisher, second son of the First Resident Commissioner of the Colony of South Australia. As in Figure 5, strong dips in the barometric pressure in the observed pressure record matched accounts of severe weather and storms in the logs.

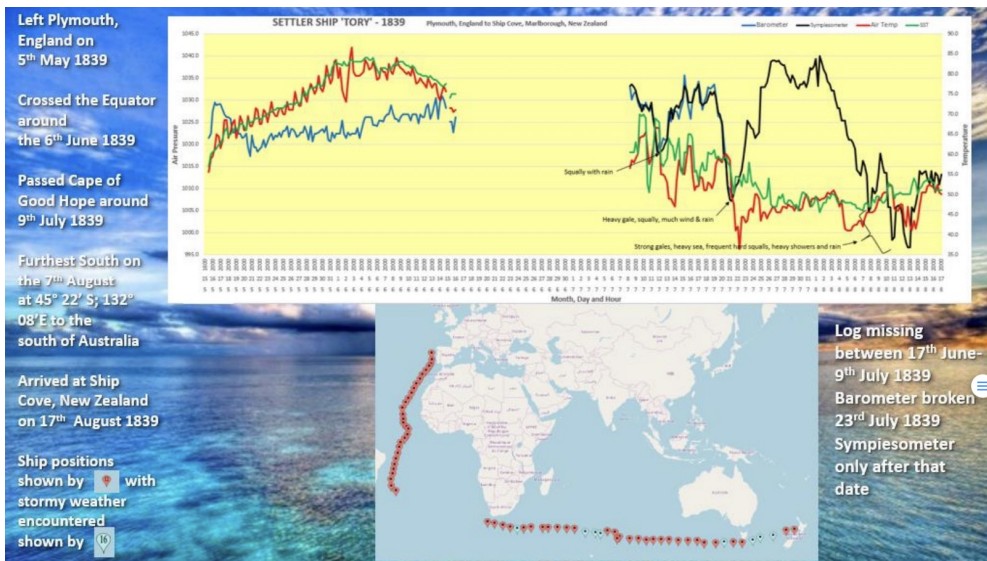

**Figure 7.** Map showing the route of the settler ship *Tory* in 1839 from England to New Zealand, with the once-daily air temperature (in red), barometric pressure (in blue) and sympiesometer (in black) observations plotted in the graph above. The sympiesometer is a type of barometer which uses oil and hydrogen gas instead of a column of mercury to measure and allow a more rapid response to air pressure changes. (https://www.wikiwand.com/en/Sympiesometer (accessed on 8 January 2024)). The observations were made by a German naturalist, Ernst Dieffenbach, traveling as a representative of the New Zealand Company. As in Figure 5, strong dips in the barometric pressure in the observed pressure record matched accounts of severe weather and storms in the logs.

## 6. The Mauritius Project: Historical Weather Observations Extracted from Ship Logbooks

The Mauritius Project, which took nearly 8 years to come to fruition, and has involved the international ACRE initiative partnering with the Meteorological Society of Mauritius

(in conjunction with the Mauritius Meteorological Services) in order to recover, scan/image, digitize, archive, and preserve old terrestrial and marine weather observations, was held in the National Archives of Mauritius and the Mauritius Meteorological Services. These are specifically as follows:

(1)    Observations extracted from ship logbooks in 188 volumes of Charles Meldrum's 'anemological' journals from 1853 to 1914.

(2)    Ship logbooks from 1848 to 1874.

(3)    Terrestrial weather observations for Mauritius, Le Réunion, Rodrigues, Seychelles, and Diego Garcia Islands (including data from Colonel Lloyd's Colonial Observatory at Port Louis) from the late 18th to the early years of the 20th century.

The 'anemological' journals have been the initial focus of the project and contain important historical ship weather observations from vessels traveling around southern Africa on the old shipping routes through Mauritius to India, China, and Australia in the period 1853 to 1914. This material also contains Indian Ocean island station records from Mauritius, Le Réunion, Rodrigues, the Seychelles, and Diego Garcia in the second half of the 19th and early 20th centuries. The collection includes ship information, location data, and a variety of meteorological parameters. These are once or twice-daily records from vessels traveling across the Indian Ocean. A later focus on the ship logbooks from 1848 to1874 will add to the above.

The scanning and digitizing effort from 2021 to 2023 was undertaken at the National Archives of Mauritius with funding from the UKMO Newton Fund Climate Science for Service Program (CSSP) China via ACRE to the Meteorological Society of Mauritius and the Mauritius Meteorological Service. A sample of the scan and digitized data from the day of the 2nd of February 1879 in the 'anemological' journals is shown in Figures 8 and 9, respectively. Note that only the instrumental weather observations on the LH side of each daily journal entry were digitized.

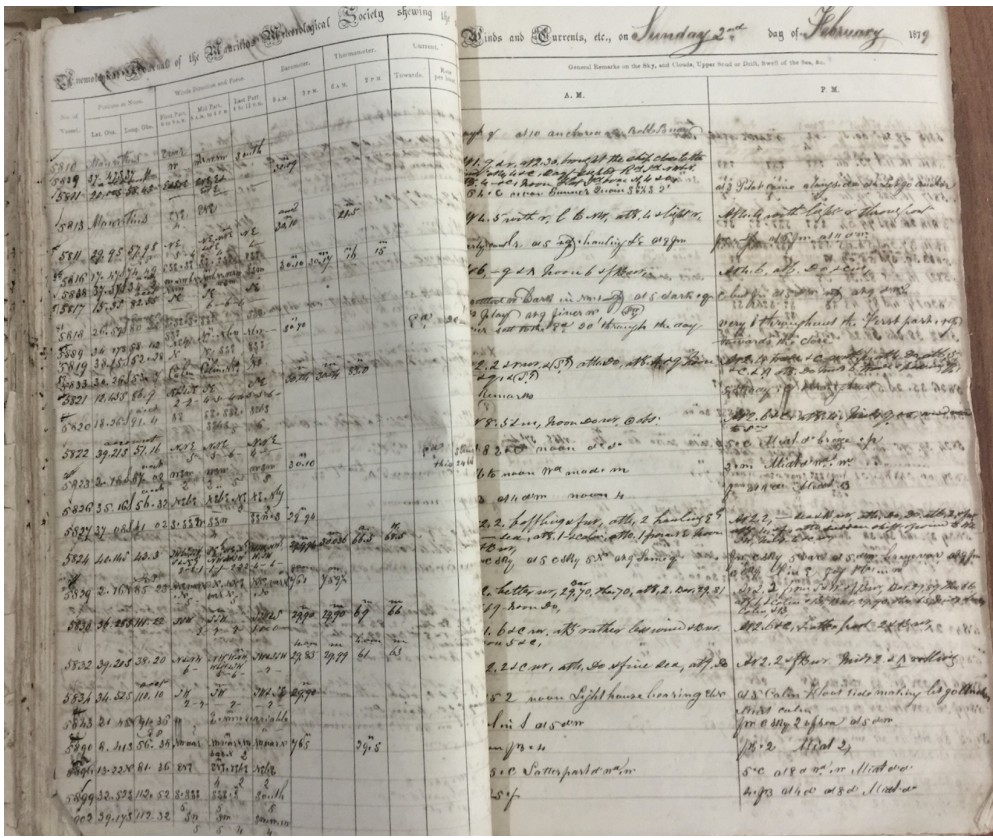

**Figure 8.** Example of a scan of the page in the 'anemological' journal for the 2nd of February 1879.

| No. of Vessel | Lat Obs | Long Obs | First Part 0 to 8am | Mid Part 8am to 4pm | Last Part 4pm to 12pm | 9am | 3pm | 6am | 2pm | Towards | Rate per Hour | Swell From |
|---|---|---|---|---|---|---|---|---|---|---|---|---|
| 5810 | Mauritius | | Evar | | | | | | | | | |
| 5839 | 37.47S | 37.4 | W | n W . WNW | South | | | | | | | |
| 5811 | accounts 20.00S | accounts 58.43 | E & ENE --- | ENE E & ENE --- | | n 30.09 | | | | | | |
| 5813 | Mauritius | | ENE 4 - | ENE -- | | | | | | | | |
| 5811 | 29.9S | 57.98 | NE 5-- 4 | n NE . NNE & NE 6-- 4-- | NE 4-- | ane n 30.10 | | n 21.5 | | | | |
| 5816 | 17.47S | EP 74.49 | ESE . SE | n SE . ESE | ESE | | | | | | | |
| 5838 | 37.37S | acct 34.2 | W & WbN NNW | n NWbN . WNW & WSW | SSW | n 30.10 | m 30.17 | n 16 | m 15 | | | |
| 5817 | 15.3S | acct 82.55 | SE -- | SE 6-- 6-- 6-- | SE -- | | | | | | | |
| 5818 | 26.59S | EP 80.22 | E . SE . S . SSE | SSE | SSE | | | | | | | |
| 5889 | 34.17S | 58.42 | NEly | n N . NbW | NbW | n 30.70 | | | | Ed | 20' | |
| 5819 | 38.15S | 52.28 | N | n N . SSE | SSE | | | | | | | |
| 5833 | 30.26S | 53.7 | Calm | n Calm . NNE | NE | | | | | | | |
| 5821 | accounts 12.45S | accounts 86.9 | SEbS & SE 2-- 2-- | SE 4--5-- 4-- 4-- | SE 5-- 5-- 6-- | n 30.04 | m 30.04 | n 83.0 | | | | |
| 5820 | 18.26S | acct 91.4 | SE | n SE . SSE . SEbS 5 | SEbS 5 | | | | | | | |
| 5822 | accounts 39.21S | accounts 51.16 | NNE 5 -- | NNE 5-- 6-- | NNE 4-- -- | | | | | | | |
| 5823 | 2.18S | acct 87.02 | WSW 2 | WSW 2    5 | WSW 5 | n 30.10 | | | | Ed | strong 24hrs | |
| 5826 | 35.16S | acct 56.32 | NEbE -- | n NEbE . NE ---    2 | NE .Nly ---  2 | | | | | | | |
| 5827 | 37.08S | 41.02 | S . SSW | SSW 4 | SSW . S | n 28.94 | | | | | | |
| 5824 | 40.14S | 43.3 | SWbS , SSW , S & SEly 2--2--1 | n SEly , NE , Nly , NW WSW --1-- 1-- 2-- 2-- 2-- | WSW . NWly , W , SW | n 29.994 | m 30.036 | a 66.5 | W 68.5 | | | |

**Figure 9.** Digitized LH page from the 'anemological' journal for the 2nd of February 1879.

Some of the data were digitized by ACRE/Copernicus Climate Change Service Data Rescue Service (C3S DRS)/UKMO Newton Fund Weather and Climate Science for Service Program (WCSSP) South Africa. With this funding, the weather observations in the 'anemological' journals in some months of 1853, and for each year from 1859 to 1900, have been scanned, digitized, and quality controlled (1876 data are still being finalized). The years 1854–1858 and 1901–1914 have yet to be completed due to the loss of funding after March 2023. The report on the project up to the end of March 2023, when the above funding finished, can be found at https://www.dropbox.com/scl/fi/vsygk3ovuiv6tqobcbmup/WCSSP_SA_End-of-Contract_Report-2023-c.docx?rlkey=iusume6qrferdw143h8674x2x&dl=0 (accessed on 8 January 2024). There is also the potential to provide considerable additional

information on the above ships using the listings of arrivals and departures of vessels at and from Port Louis on Mauritius in monthly tabulations in the Mauritian newspapers of the time. These detail ship names, nationality, tonnage, captain's name, arrival date, where from, cargo, agents, departure date, where bound, cargo, agents, and observations when in harbor (e.g., loading).

The great bulk of vessels detailed in the above newspapers were from the old colonial powers in Europe, particularly Britain and France. There were then ships from the United States, Germany, Sweden, Denmark, the Netherlands, Norway, Italy, Austria, and Russia. As the weather observations made on these vessels have been extracted from their logbooks, and are from merchant ships, it is mostly highly likely that the majority of the original logbooks may not have survived. They are thus a very important source of historical data at these times and in the regions that they traversed.

In their study of historical tropical cyclones in the Asian region centering on Japan, [27] noted that European or American ships sailing along Asian coasts before the mid-nineteenth century were the only sources of instrumental meteorological observations. When these ships anchored at a port, a number of them continued measuring the weather and, if stationed in such ports for weeks, months, or years, they were vital sources of longer-term data for reconstructing past weather patterns. This is especially true for our understanding of historical severe weather events like tropical cyclones. One particularly interesting finding in preliminary investigations of the digitized data, that was gleaned in conjunction with an examination of the cargo listed for each vessel in the Mauritian newspapers during the 1870s period, were ships sailing to the wider Indian Ocean, with a stop in Mauritius, which were involved in the Guano trade. There were some 50–60 'Guano' vessels identified in this initial probing of the 1870s portion of the data set, that sailed from South America to Mauritius, traveling around Cape Horn across the southern Atlantic, then around the Cape of Good Hope and South Africa. The portion of their route across the southern Atlantic Ocean is unlikely to have been traversed by any other vessels in a quasi-routine manner in such a period, making the observations made on such voyages extremely valuable in filling a significant gap in the data coverage at these times. This can be seen in the two examples shown in Figure 10 for January to February 1871 and in Figure 11 for July to October 1871, where each vessel's passage is displayed on each map along with a plot of the daily air temperature and barometric pressure observations in the bottom LH side of each diagram. Passages of this nature at such mid to high latitudes around Cape Horn and the South Atlantic during the Southern Hemisphere summer would have been taxing on the ship and crew but doing so during the Southern Hemisphere winter would have been outright precarious. The time taken to make these similar voyages in distance is also indicative of open ocean weather conditions in each season; during the summer, the passage took just short of 2 months (55 days), while during the winter the passage lasted over 2 and a half months (68 days). This work on the Guano ships will be extended to investigate such vessels in the full 1853–1914 journal database.

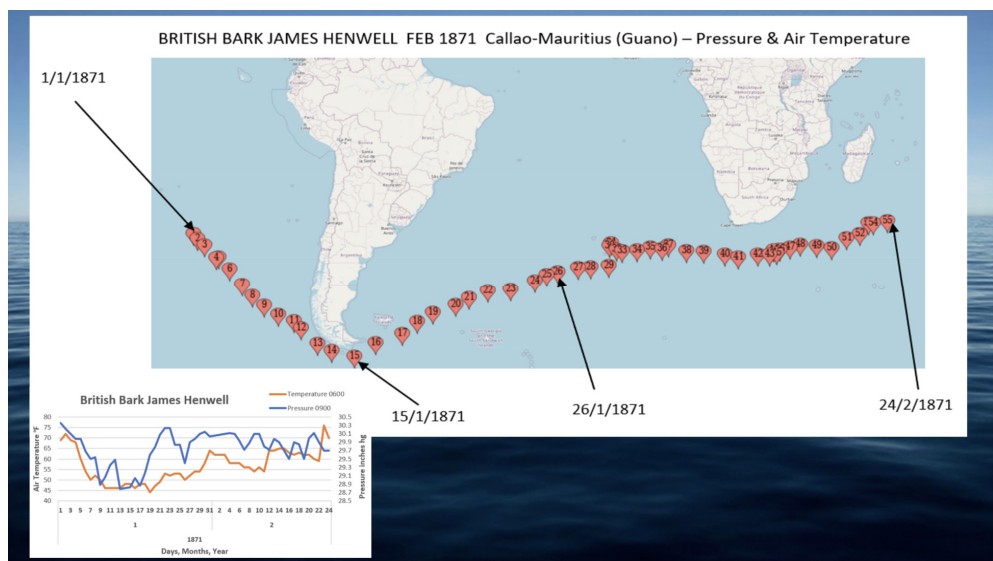

**Figure 10.** The voyage of the British bark *James Henwell* from Callao, Peru to Mauritius in January to February 1871. The vessel's route is shown on each map along with a plot of the daily air temperature and barometric pressure in the bottom LH side of each diagram.

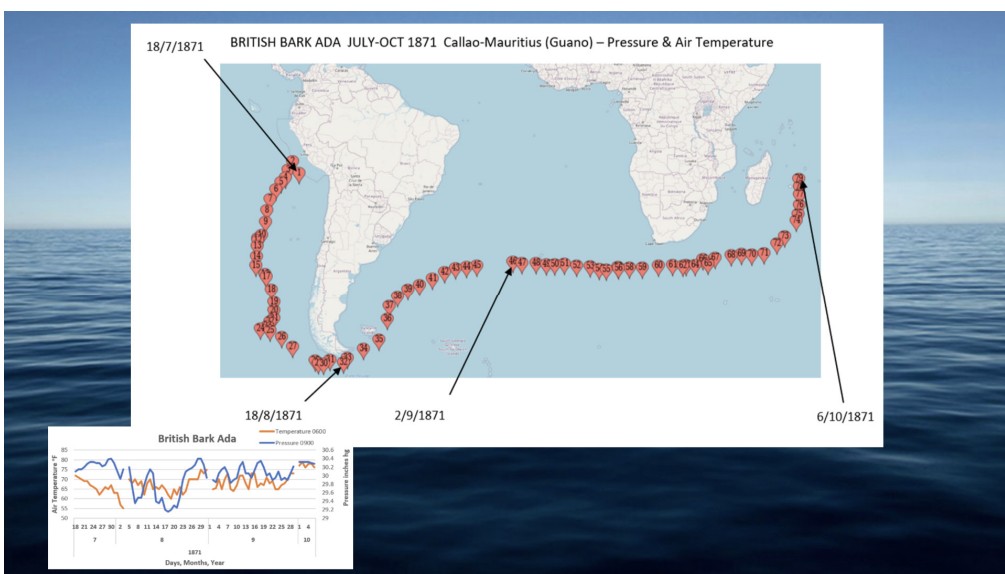

**Figure 11.** The voyage of the British bark *Ada* from Callao, Peru to Mauritius from July to October 1871.The vessel's route is shown on each map along with a plot of the daily air temperature and barometric pressure in the bottom LH side of each diagram.

### 6.1. Selected Further Ship Logs

We selected logbooks from the maritime archive collections of The National Archives (Kew, Richmond), The National Meteorological Archive (Exeter), The UK Hydrographic Office (Taunton), The Institute of Maritime History at Åbo Akademi University (Turku), and The Åland Maritime Museum (Mariehamn).

A plethora of logbooks from ships of the Royal Navy in the 19th century can be found in the maritime archive collections of (a) The National Archives in Kew, Richmond, (b) The National Meteorological Archive in Exeter, and (c) The UK Hydrographic Office in Taunton. These collections include a variety of ships' logbooks, weather books, meteorological registers, private weather diaries, composite and individual remark books, and miscellaneous papers. The ACRE/UKMO Newton Fund Weather and Climate Science for Service Program (WCSSP) South Africa facilitated the preservation of these archives

with the scanning/imaging and digitization of the aforementioned logbooks, as well as the quality control of the digitized data. These logbooks cover the following time periods:

(1)   The National Archives (one-hundred and thirty-four completed logbooks)—from 1832 to 1833, from 1853 to 1880 and from 1898 to 1899;
(2)   The National Meteorological Archive (seven completed logbooks)—from 1849 to 1882;
(3)   The UK Hydrographic Office (forty-six completed logbooks)—1816, from 1823 to 1825 and from 1844 to 1868.

However, there are nine logbooks from The National Archives (years 1856–1857, 1863–1866 and 1899–1901), six logbooks from The National Meteorological Archive (years 1856–1857, 1862, 1867–1868 and 1891–1892) and twenty-four logbooks from The UK Hydrographic Office (years 1862–1865) that have not been completed due to the loss of funding after March 2023.

There has also been important imaging of logbooks from Norway, plus work on Chilean Navy logbooks, while there are currently ongoing initiatives with the Argentine Navy, though this information may be sensitive. In addition, the German Weather Service (DWD) is continuing to image and key their entire collection of German meteorological logbooks. Thus, marine data rescue initiatives are actively pursuing the use of non-English language ship logbooks.

Additionally, there is also an extensive archive of Finnish logbooks (written in Swedish) derived from The Institute of Maritime History at Åbo Akademi University in Turku and The Åland Maritime Museum in Mariehamn. Some of these logbooks have also been scanned/imaged and digitized:

(1)   The Institute of Maritime History at Åbo Akademi University—fifteen completed logbooks from 1850 to 1899 and three remaining logbooks (years 1862–1863, 1876–1877 and 1899–1901);
(2)   The Åland Maritime Museum—two completed logbooks (1853 and from 1880 to 1882).

These ships traveled from England to South Africa, China, Japan, Philippines, and Malaysia, as well as from Finland to South Africa. The duration of the voyages lasted from several months up to three years. During traveling, the vessels' crew recorded daily route information (longitude–latitude), remarks regarding the ship and the voyage (employment, deaths on board, ship damages, and maintenance, etc.), meteorological parameters, observed weather, and other events. However, the handwritten nature of the logbooks (calligraphy and different writings in the same logbook) made the recordings hardly readable. The meteorological observations usually refer to wind (speed and direction), barometric pressure, and air and sea temperature. During sailing, the meteorological observations were performed hourly or every few hours, while when at anchor the observations were performed every two hours. Figures 12–15 are examples of the vessel *HMS Argus* (The National Archives) that cruised in 1869 from Japan to England.

*6.2. Old Weather WW2 and Weather Rescue at Sea*

Two projects which used citizen science to recover millions of marine weather observations are now discussed. Old Weather WW2 rescued historical weather observations from United States Navy (USN) ships during World War 2 (WW2), and Weather Rescue at Sea (WRS) used UK naval logbooks to fill the gap in observational datasets in the 1860s. Both projects harnessed the cumulative power of crowd-sourced transcription to data-rescue historical observations.

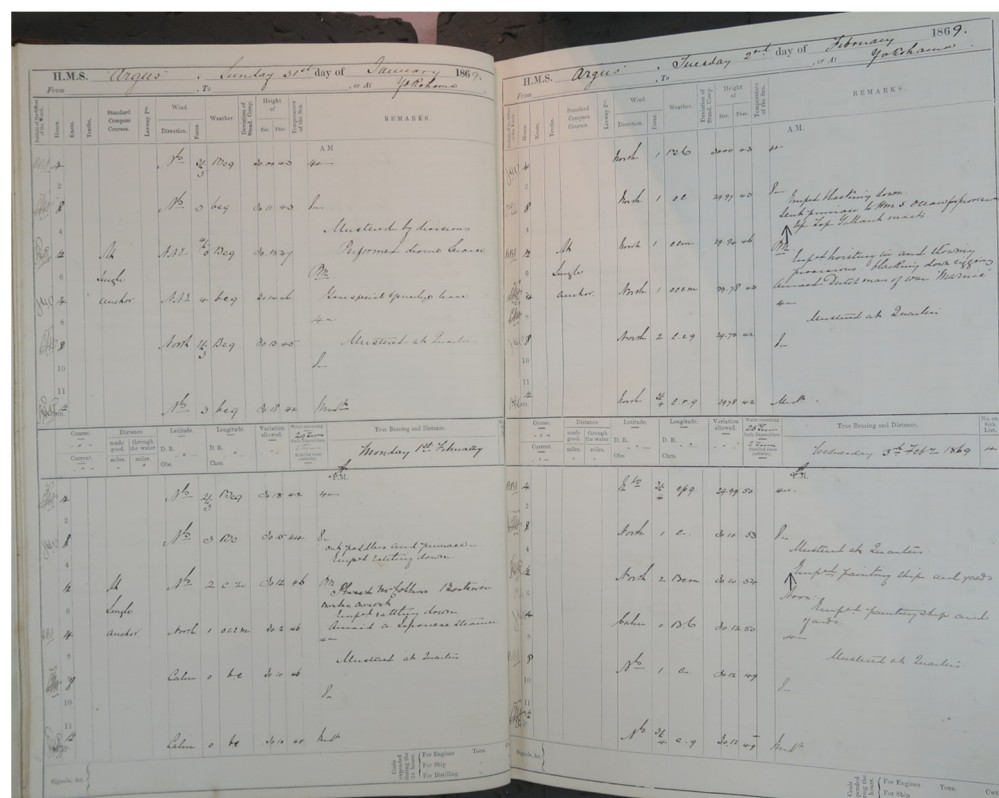

**Figure 12.** Example of two scanned pages from *HMS Argus* logbook. These pages refer to four days (from 31 January to 3 February 1869). The vessel is moored in Yokohama.

| Year | Month | Day | Hour | Wind direction | Wind force | Weather | Barometer | Thermometer | Sea Temperature | D.R. Lat. | Obs. Lat. | D.R. Lon. | Chro. Lon. |
|---|---|---|---|---|---|---|---|---|---|---|---|---|---|
| 1869 | 1 | 25 | 1 a.m. | SE by E | 2 | bc | | | | | | | |
| 1869 | 1 | 25 | 2 a.m. | SE by E | 2 | | | | | | | | |
| 1869 | 1 | 25 | 3 a.m. | ESE | 2 | oc | | | | | | | |
| 1869 | 1 | 25 | 4 a.m. | SE | 2 | cmp | 30.10 | 55 | | | | | |
| 1869 | 1 | 25 | 5 a.m. | SE | 2 | cp | | | | | | | |
| 1869 | 1 | 25 | 6 a.m. | SE | 2 | c | | | | | | | |
| 1869 | 1 | 25 | 7 a.m. | SE | 2 | c | | | | | | | |
| 1869 | 1 | 25 | 8 a.m. | Variable | 1 | c | 30.10 | 59 | | | | | |
| 1869 | 1 | 25 | 9 a.m. | Variable | 1 | c | | | | | | | |
| 1869 | 1 | 25 | 10 a.m. | NW | 1 | c | | | | | | | |
| 1869 | 1 | 25 | 11 a.m. | NW | 1 | c | | | | | | | |
| 1869 | 1 | 25 | 12 a.m. | NW | 2 | c | 30.12 | 60 | | | | | |
| 1869 | 1 | 25 | 1 p.m. | NW | 2 | c | | | | | | | |
| 1869 | 1 | 25 | 2 p.m. | NW | 2 | c | | | | | | | |
| 1869 | 1 | 25 | 3 p.m. | NW | 2 | cp | | | | | | | |
| 1869 | 1 | 25 | 4 p.m. | NW | 1 | cr | 30.07 | 65 | | | | | |
| 1869 | 1 | 25 | 5 p.m. | Calm | 0 | cmr | | | | | | | |
| 1869 | 1 | 25 | 6 p.m. | Calm | 0 | cmr | | | | | | | |
| 1869 | 1 | 25 | 7 p.m. | NNW | 3 | cmr | | | | | | | |
| 1869 | 1 | 25 | 8 p.m. | NNW | 4 to 5 | cmr | 30.03 | 53 | | | | | |
| 1869 | 1 | 25 | 9 p.m. | NNW | 4 to 5 | cmr | | | | | | | |
| 1869 | 1 | 25 | 10 p.m. | NNW | 2 to 3 | cmqr | | | | | | | |
| 1869 | 1 | 25 | 11 p.m. | NNW | 2 to 3 | cmqr | | | | | | | |
| 1869 | 1 | 25 | 12 p.m. | NNW | | cmqr | 29.99 | 56 | | | | | |
| 1869 | 1 | 26 | 1 a.m. | N | 3 to 5 | cqr | | | | | | | |
| 1869 | 1 | 26 | 2 a.m. | N | 3 to 5 | cqr | | | | | | | |
| 1869 | 1 | 26 | 3 a.m. | N | 3 to 5 | cqr | | | | | | | |
| 1869 | 1 | 26 | 4 a.m. | N | 3 to 5 | cqr | 29.88 | 50 | | | | | |
| 1869 | 1 | 26 | 5 a.m. | N | 4 to 5 | cqr | | | | | | | |
| 1869 | 1 | 26 | 6 a.m. | N | 4 to 5 | cqr | | | | | | | |
| 1869 | 1 | 26 | 7 a.m. | N | 3 to 4 | cqr | | | | | | | |

**Figure 13.** Digitized page from *HMS Argus* logbook from January 1869.

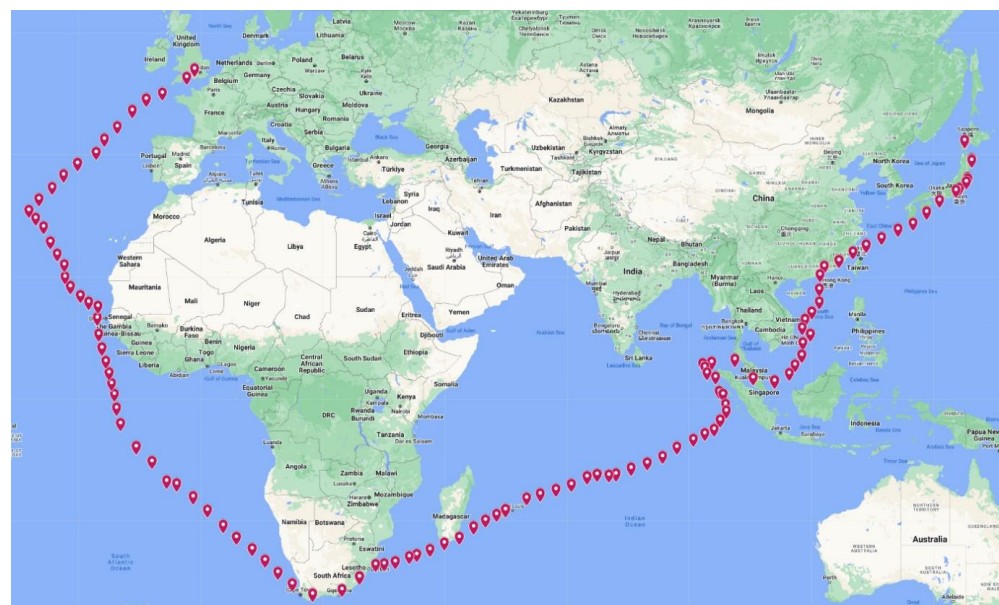

**Figure 14.** The voyage of the Royal Navy ship *HMS Argus* from Hakodadi, Japan to Portsmouth, England from January to December 1869.

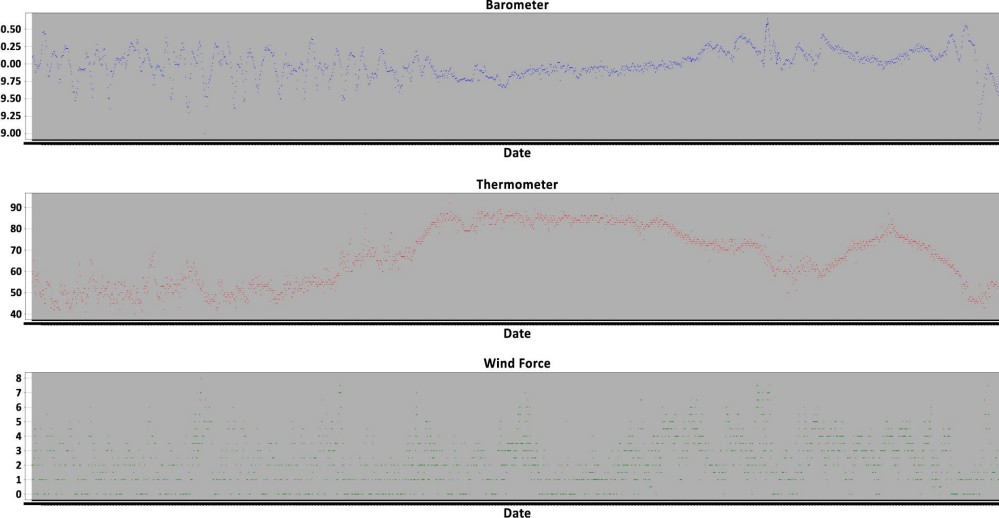

**Figure 15.** Plots of barometric pressure (**upper**), air temperature (**middle**), and wind speed (**bottom**) during the voyage of *HMS Argus* in 1869.

*6.3. Old Weather WW2*

All climate reconstructions show that the global oceans have warmed since the start of the 20th century, but there is anomalous warmth in global mean SSTs during the WW2 period (between 1941 and 1945) when compared to the preceding and following 5-year periods [28]. Also, the uncertainty in the estimated anomaly for this period is several times larger than for more recent periods.

Several possible explanations have been put forward to account for this anomaly, referred to as the WW2 warm anomaly (WW2WA) by previous studies, such as the reduced number of observations [28,29] and changes in the types of SST measurement [30,31]. When WW2 commenced, trade routes were severely disrupted, limiting observations taken by voluntary observing merchant ships (VOS) which usually criss-crossed the global oceans. This caused a large drop (58%; [29] in the number of marine observations available for the duration of WW2.

More crucially, poorly documented changes in the observing practices may have led to large biases and errors. For example, the preference for taking SST measurements from the inlet water pipes used to cool engines (known as engine room intake, ERI), in contrast to hauling canvas/wooden buckets onboard, resulted in a warm bias in the aggregated SSTs [32]. The rapid rate of these transitions is not always well documented and can be mislabeled, which impedes the correct adjustments being applied to the observations [28]. Another practice changed during WW2 was that more observations were taken during daytime than night-time. Both of the above changes are assumed to be due to the need to reduce exposure to the enemy ships and avoid being detected [28,33]. Without additional data and documentation of prevailing practices, disentangling the reasons for the WW2WA is very difficult.

Most of the marine observations taken during WW2 were on board naval ships of various countries. However, many observations were destroyed as an act of war, or simply forgotten due to the length of time they were considered classified. To fill gaps in observational coverage and contribute to improving metadata regarding observing practices, the NOAA-funded project 'Old Weather: World War 2′ gathered thousands of volunteers to transcribe weather observations from logbooks of US destroyers and other naval ships which were part of the US Pacific fleet based at Hawaii. These ships saw action in the Indo-Pacific and Far-East including the Pearl Harbor attack, taking observations at times and places where few or no other digitized observations exist.

In 2017, the National Declassification Center (NDC) at the National Archives and Records Administration (NARA) released nearly 200,000 pages of formerly classified U.S. Navy Command Files from the WW2 era. The files consisted primarily of records from the Pacific Theatre between 1941 and 1946. The files contain many kinds of documents, maps, ship logbooks, photographs, etc. Here, we focus on the ship logbooks containing meteorological observations (Figure 16).

A dataset of more than 3.7 million observations has been rescued [34]. The dataset has more than 630,000 unique records, where each record contains the date and time, positional information, and one dry-bulb temperature (Tdry), wet-bulb temperature (Twet), Twater (SST = sea–surface temperature), barometer-attached thermometer temperature (Baro At. Therm.), and pressure observation. There are 611,223 observations of air pressure, 197,716 observations of Baro At. therm., 601,978 observations of dry bulb temperature (Tdry), 604,155 observations of wet bulb temperature (Twet), and 314,713 observations of SST. There are an average of 7000 records per ship per year, and each ship logbook has observations for around 300 days per year on average. All ship tracks are supported by documentary evidence about the ships' movements from other sources [35]. Over the 5-year period, the various ships traveled across the Pacific, Indian, and Atlantic oceans, providing a rich dataset all across the globe (Figure 17).

As an example of the data available, Figure 18 shows the track of *USS Pennsylvania* during the 1941–1945 period. During 1941 and 1942, the ship traveled between San Francisco and Pearl Harbor. In 1943, it made trips to the Aleutian Islands near Alaska, Marshall Islands, and Guam in the Pacific. For the year 1944, meteorological observations are present, but navigation data is missing; hence, the year is empty. In 1945, it traveled to Papua New Guinea and Philippines and other islands in the South China Sea from Pearl Harbor. Then, it reached Puget Sound Naval Shipyard in Washington towards the end of 1945. The meteorological observations of pressure and Tdry closely reflect the regions traveled.

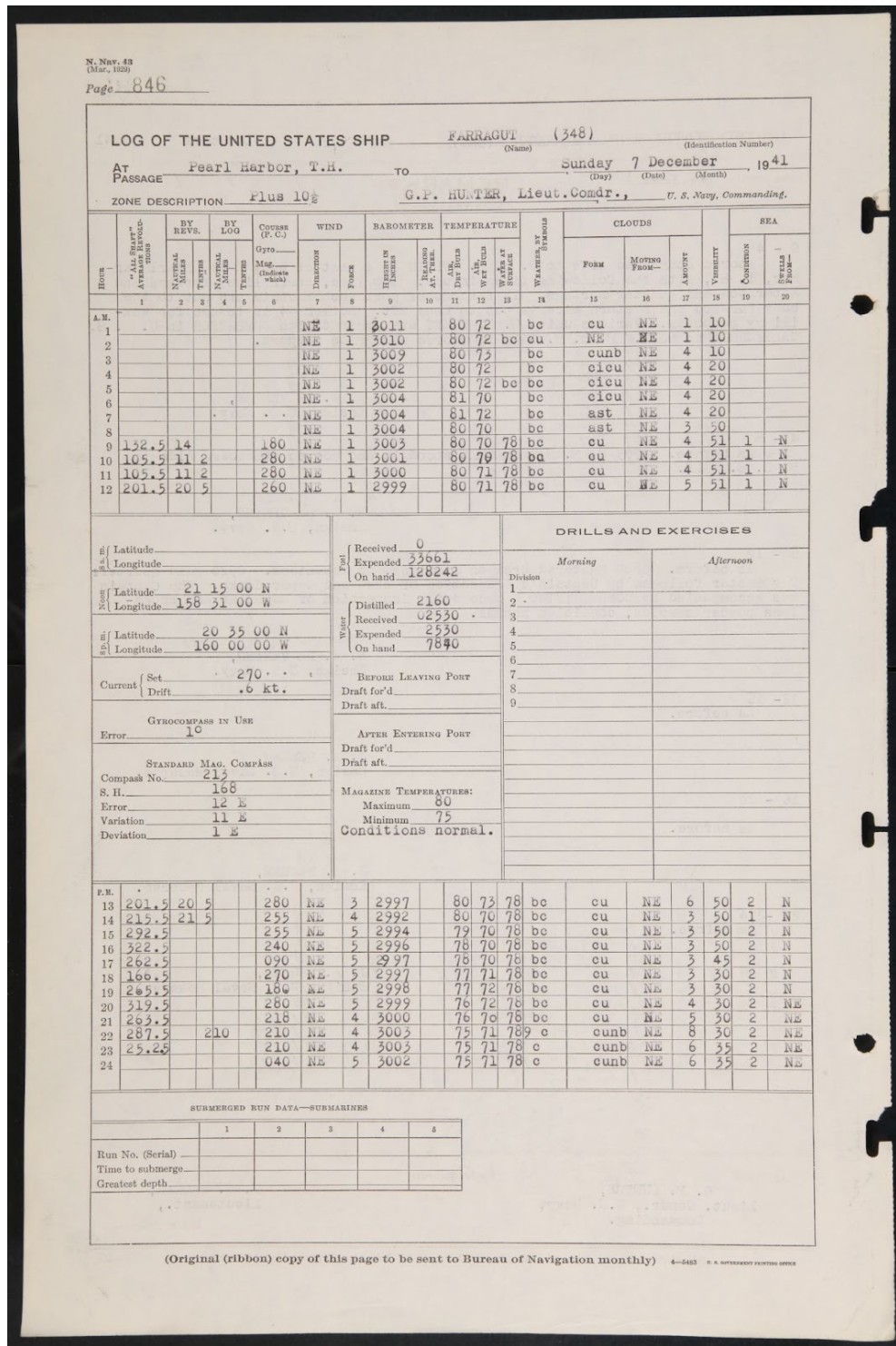

**Figure 16.** A typical US Naval ship logbook 'Navigation & Observations' page used during WW2. Information about the ships' name, passage to/from, date, zone, and commanding officer are noted at the top. Meteorological and navigation information is recorded in their respective columns. This page is from USS Farragut on the day of the attack on Pearl Harbor (7 December 1941) (Source: [34]).

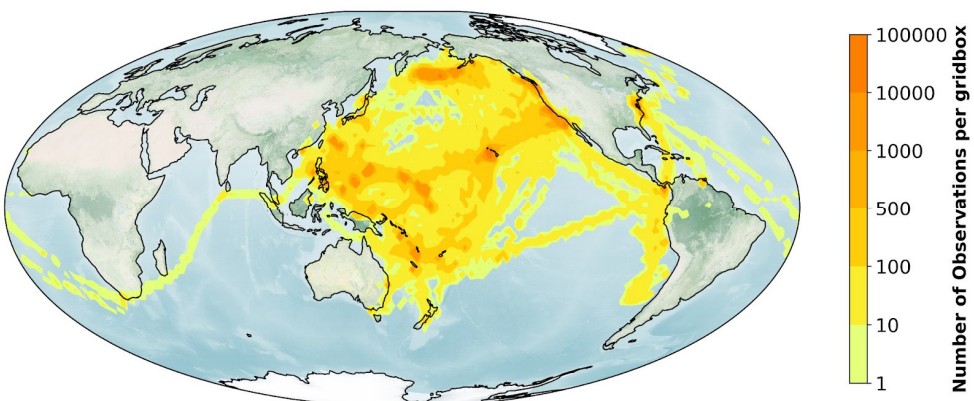

**Figure 17.** Observational density of all observations (1941–1945) in the dataset binned into a $2° \times 2°$ regular grid.

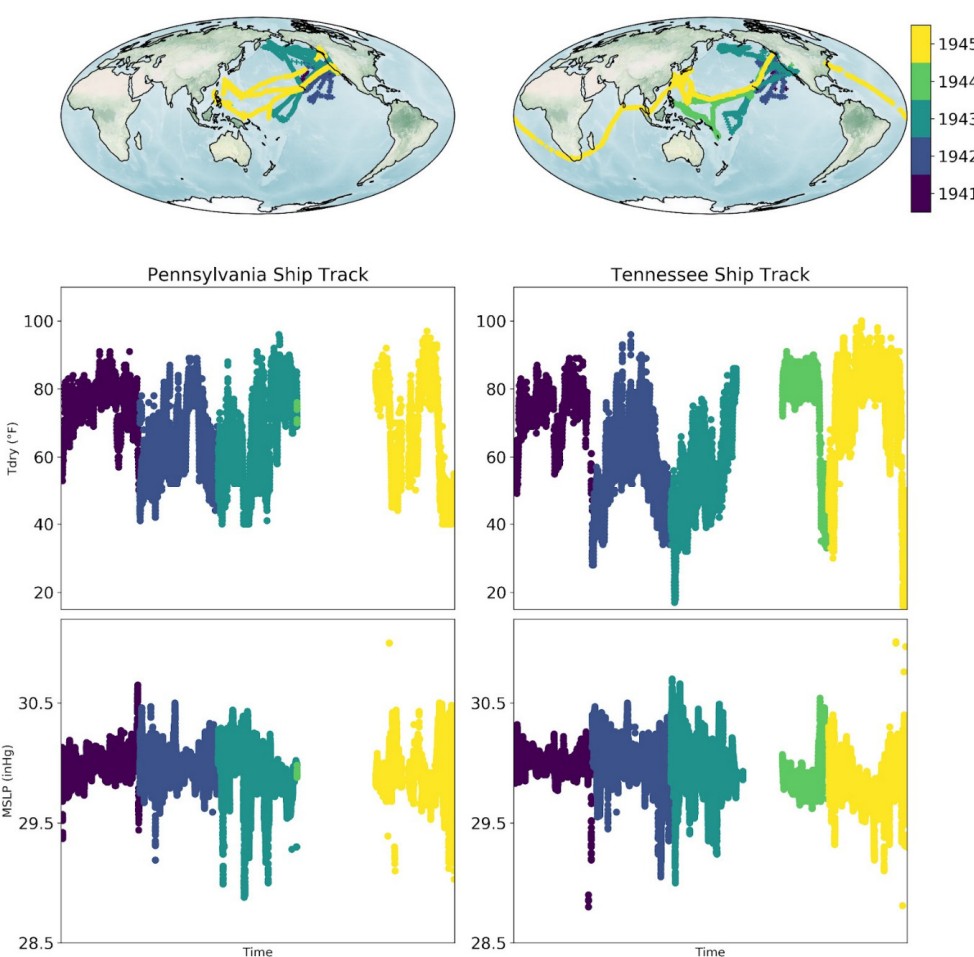

**Figure 18.** Ship tracks of *USSs Pennsylvania* (**left**) and *Tennessee* (**right**), including MAT and sea level pressure (SLP) observations during the 1941–1945 period (Source: [34]).

Figure 18 also shows the track of *USS Tennessee* over the 1941–1945 period. During 1941, the ship traveled to Pearl Harbor from San Francisco, reaching Puget Sound Naval Shipyard in Washington at the end of the year. 1942 was spent completing various exercises off California and in the seas around Hawaii. The years 1943, 1944, and 1945 were long-distance trips, first to Aleutian Islands, then Fiji, Marshall Islands, and Philippines. In 1945, it started from the Naval Shipyard in Washington and traveled to the southern coast of Japan via Hawaii, and also included multiple trips to the Chinese coast. Starting from

Japan, the ship then visited Taiwan, Singapore, Sri Lanka, Cape Town, finally reaching New York, completing a circumnavigation.

Several studies have highlighted severe dust droughts and heat waves in North America during the 1930s, followed by a strong 1939–1942 El Niño event which had a significant impact over the globe. The El Niño during 1939–1942 led to extremes in global climate anomalies, including cold winters in Europe, warm winters in Alaska, wet springs in central Europe, and a drought in Australia. However, our understanding is only partially complete due to the severely limited coverage of observations for the WW2 period; the presented dataset in this study can help fill in some of the gaps.

*6.4. Weather Rescue at Sea*

Observing and following the weather through the changing seasons was crucial to survival in the pre-industrial era. It was more so for those who spent long periods of time onboard ships traveling across the globe. In the age of sail, knowledge of winds and currents was crucial to reach their destinations safely and on time. Out of practical necessity, gradually, maritime nations developed several weather observing instruments and procedures to record the weather encountered on long sea journeys. And, in 1854, a maritime conference of sea-faring nations tried to codify observational taking and record keeping helping to standardize and share observations among themselves [36]. That process amassed an enormous number of 'standard' logbooks containing detailed sub-daily weather observations at sea from around the globe.

There is a strong scientific interest in understanding the climate of the early industrial era against which our present climate could be measured, to assess anthropogenic impact on climate change. As large parts of the globe are covered in ocean, many previous studies have used historical marine observations to estimate these changes in the climate. The CLIWOC project [37], a multinational study, systematically collected, extracted, and analyzed UK–Spain–Dutch ship logbooks before 1850. Brohan et al. [38] produced a substantial number of historical data from English East India Company ship logbooks starting from 1789 and ending in 1834. They produced more than 200,000 records containing three meteorological variables (temperature, pressure, and wind), giving unique insight into historical climate. This study provided further evidence that historical ship logbook observations can be used to study climate variability when land-based observational networks are not dense enough.

To further the development of the reconstruction of past climate by enhancing the data available to them, the international ACRE initiative [39] coordinates various data-rescue efforts and communities. One of the narrowest bottlenecks of historical data extraction has been a lack of reliable and efficient automated processes to deal with hundreds of thousands of weather journals and ship logbooks which are written by hand. Many new archives have been located, cataloged, and photographed by the data-rescue initiatives. However, there is at least as much data to be rescued as are currently available in digital archives for the period prior to 1950 [25].

Data rescue (transcribing hand-written observations into computer-readable digital format) of historical logbooks has been taking place for decades, but manually transcribing an almost inexhaustible number of logbooks by individual researchers would take thousands of human lifetimes. As a result, large gaps have remained in our knowledge of the climate, both in space and time. The 19th century has fewer observations available than the 20th century in the world's largest observation meteorological dataset, ICOADS version 3 (International Comprehensive Ocean-Atmosphere Data Set, [29]). On closer inspection, the average number of monthly observations and percent of global coverage in the 1860s and 1870s is relatively poor compared to other decades after 1850.

For the volume of data contained in the collection described here, a traditional manual transcription approach would have taken many person-years of effort. Instead, the availability of scanned images of the ship logbooks enabled the creation of a science project that asked volunteers to transcribe the observations into digital form more efficiently.

The Zooniverse platform (www.zooniverse.org (accessed on 8 January 2024)) offers a flexible framework upon which various citizen science projects have been built. Many different themes are represented on the platform, from astronomy, biology, ecology, and conservation to historical documents. The original Old Weather project was one of the first projects to extract historical weather observations contained in ship logbooks from an extended period around WW1. Since then, many projects have successfully used Zooniverse to digitize historical weather observations, e.g., WeatherRescue.org [40,41], RainfallRescue.org [42], SouthernWeatherDiscovery.org [43], Climate History Australia [44], and Meteorologum ad Extremum Terrae [45].

Within this context, the Weather Rescue At Sea (WRS) project has used the citizen science based Zooniverse platform to recover some of these observations and make them usable, with a focus on ships traveling through the Atlantic, Indian, and Pacific ocean basins in the 1860s and 1870s. The focus has been on logbooks archived at the UKHO (UK Hydrographic Office) that are best suited to produce data in the targeted time period with global coverage (Figure 19). Filling in the gaps in our knowledge will remove ambiguity in how the climate varied historically in many regions where observations are currently poor or non-existent. The data generated through this project will also help to fill many crucial gaps in the large climate datasets (e.g., ICOADS) which will be used to generate new estimates of the industrial and pre-industrial era baseline climate. But more generally, this data and data from other historical sources are currently used to improve the models and reanalysis systems used for climate and weather research.

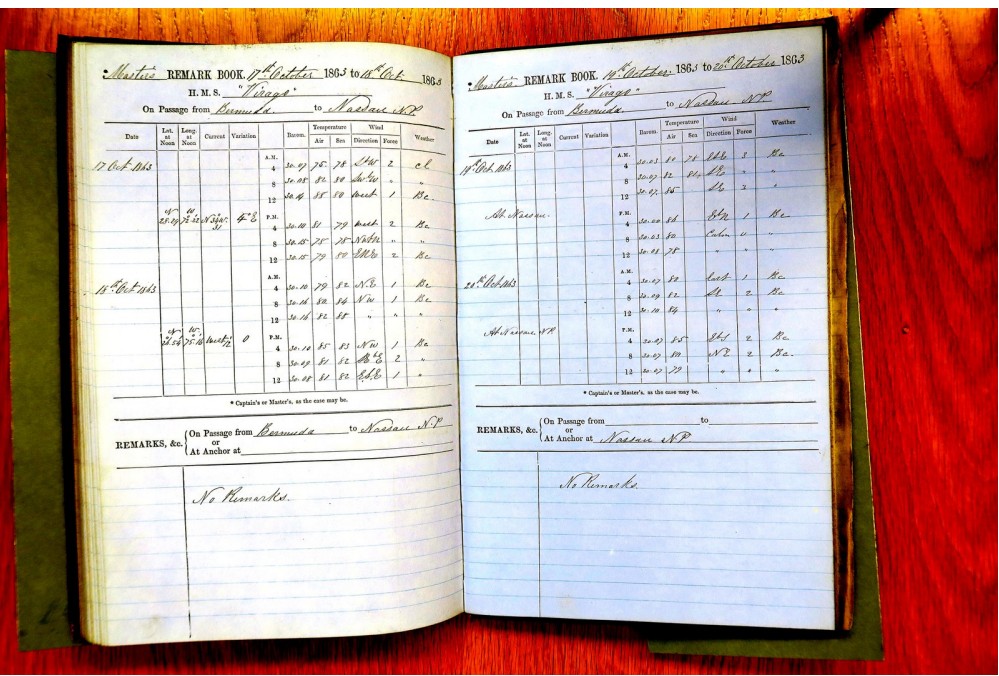

**Figure 19.** An example of a logbook page used in the WRS project. The page is taken from *HMS Virago* showing date, position, and meteorological observations.

So far, a total of 248 logbooks have been used in the project, totaling 25,000 images covering the 1860s and 1870s. More than 3000 volunteers contributed to the transcription process; the post-processing work of error corrections and consensus checking is still ongoing. So far, we have processed ~44,000 records containing navigational and meteorological observations. Figure 20 shows a snapshot of all ship tracks processed so far.

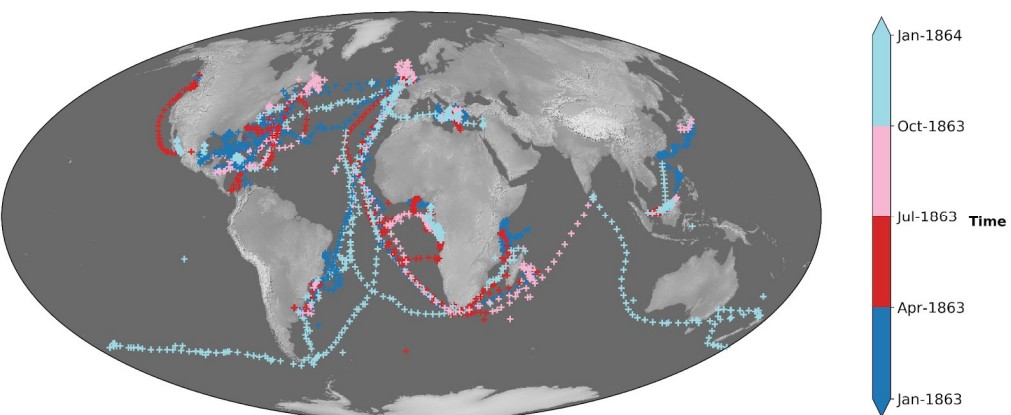

**Figure 20.** The ship tracks of all ships, commencing in January 1863 and ending in January 1864. Colors denote the timeline in the color bar.

Finally, we highlight two of the main lessons learned from both the above Old Weather WW2 and WRS projects. Firstly, the design of transcription workflows should reflect the structure of the logbook page. Providing context about the logbook pages, the purpose of the project, and where the data would be used, all helped to motivate the volunteers. Secondly, information requiring transcription should be grouped together into workflows, e.g., positions, zones, dates, and particular weather types (see [33]).

## 7. National Institute of Water and Atmospheric Research (NIWA) Activities

Initial work undertaken under ACRE focused on identifying marine observations from ships to corroborate early land-based pressure observations in New Zealand [46]. Subsequently, support through the Deep South National Science Challenge was used to identify ship-based weather observations for the region south of New Zealand to Antarctica across the Southern Ocean that would improve the 20th Century Reanalysis. This work was undertaken in a project called Southern Weather Discovery [43], which had a primary focus of setting up data transcription workflows established by other leading data rescue projects (e.g., Weather Rescue), evaluating the efficacy of AI for data transcription, and determining optimal data keying replication standards for quality control. The latest data rescue efforts in New Zealand are currently focused on two fronts; securing digital surrogates of formal observation forms in the National Institute of Water and Atmospheric Research (NIWA) archives and the recovery of ship log observations recorded on sub-daily synoptic weather maps compiled by the New Zealand Meteorological Service. The work at the NIWA archives was initiated in 2009 and is nearly completed, and the main document targets are first-class climatological stations and third-class manual rainfall observation stations. The former is the highest priority, as the sheets contain essential climate variables that are regularly reported on in terms of extremes and trends. Pressure observations from the first-class climatological stations have been aperiodically supplied to the International Surface Pressure Databank (ISPD) via the ACRE Pacific chapter. An exchange of materials held in the UKMO (United Kingdom Meteorological Office) archives has recently helped to fill time gaps for the earliest official record in Auckland, which will overlap with several ships, including the HMS Pandora, that undertook the first hydrographic survey of the colony. Critically, missing data sheets supplied by ACRE are now being used to test the validity of reported low-pressure observations related to 19th century storms believed to be of ex-tropical origin, giving a longer historical context for the recent impacts of the tropical cyclone Gabrielle. The marine observations from this time period will be valuable for gap filling the land-based observation record, which becomes more robust from the 1870s onward.

There are limited reports from ship logbooks that can provide a wider spatial context for the origins of former storms that impacted the southern mid latitudes. However, a recent trove of historical sub daily synoptic maps for New Zealand and the surrounding

oceans were transferred from NIWA to the National Archives of New Zealand in Mangere, Auckland. These historical maps are in oversize format and bound in leather, requiring digital photography for capture. They are important because they contain observations that were sent by wireless telegraph from ships to the mainland, are not likely to be found in other sources, and therefore the most current extended reanalyses without radiosondes. The main interest in obtaining these historical maps, and the marine weather observations on them, is to evaluate the occurrence and origin of storms that were characterized by deep low pressures that impacted New Zealand prior to the 1940s.

*Examples of Historical Marine Data Assimilated into Reanalyses*

The value of non-digitized marine data for producing reanalyses was demonstrated in two UKMO-Newton Fund WCSSP projects which aimed at digitizing marine data in the Southern Hemisphere during two climatically important periods. The first project targeted the 1876–1878 period (Figure 21, top), which was characterized by a very strong El Niño and was arguably one of the deadliest climate events in history. The project digitized climate data from the logs of 20 ships that cruised the South Atlantic and Indian Ocean during these years [47]). Note the many curved ship tracks, which are typical for sailing ships and provide a different coverage than the steamships, that are on a more linear track.

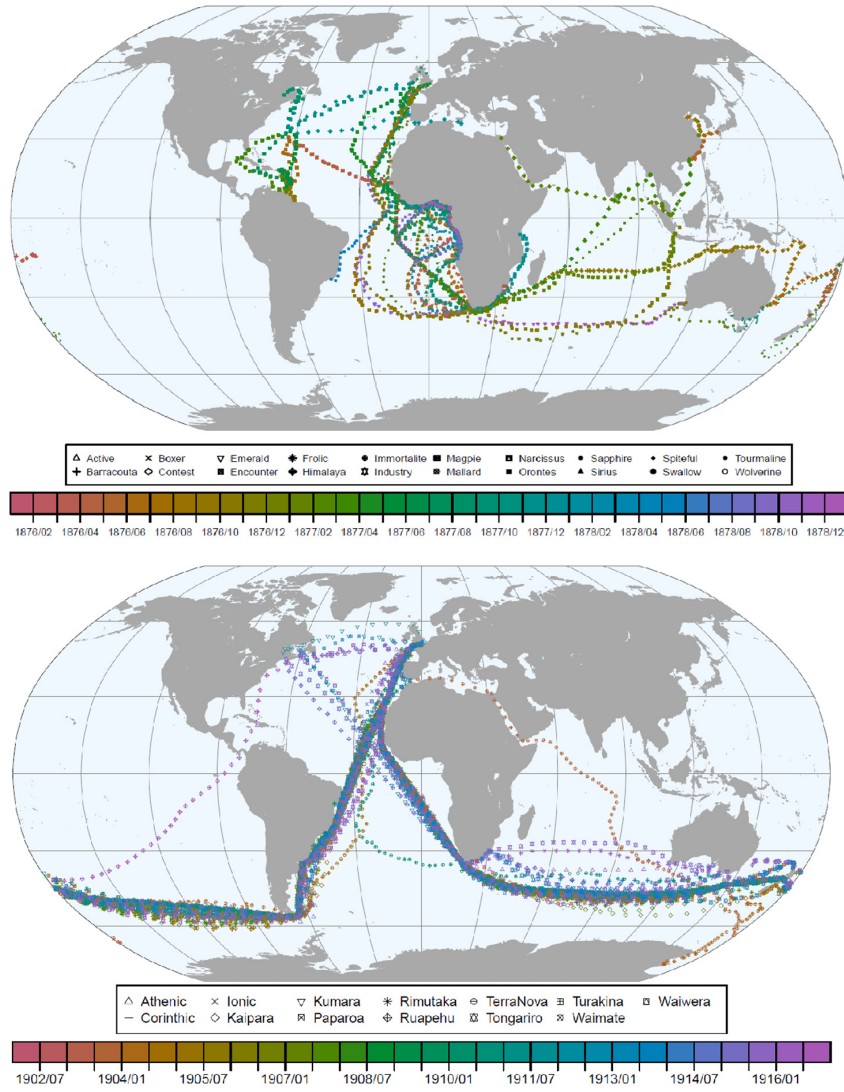

**Figure 21.** Map of the tracks of the ships for which data were digitized for (**top**) 1876–1878 and (**bottom**) 1902–1916, colored by year. Coordinates are shown for 12 UTC, when they were measured (from [47,48]).

Using an offline data assimilation approach, it was then shown that assimilating these data into 20CRv3 would increase the skill of the product [47]. Not only did correlations with independent data increase, but a storm that was not well represented in 20CRv3 appeared very clearly after assimilating the additional records. Although the added value of a single ship may seem small, the result showed a measurable improvement in the pressure fields, which in an online approach (i.e., cycling the field back to the next forecast step) is arguably even larger.

The second project [48] targeted a period around 1910, during which global temperature reached a decadal minimum. The causes of this anomaly are not well understood. Digitizing data from 13 ships in the period 1902–1916 (Figure 21, bottom) and assimilating them offline into 20CRv3 again increased the correlations and led to a reduction of the ensemble spread. The augmented reanalysis contributes to a better understanding of this anomalous period, although many questions remain open (note the more linear tracks of the steamships during these years; also, this period still saw a good coverage of the Southern Ocean, which changed rapidly after 1914 with the opening of the Panama Canal).

The data confirmed that the cold period is not an artifact of biased ship data but must be understood as an unusual combination of external factors (volcanic eruption of Santa Maria in 1902, perhaps Novarupta in 1911) and internal variability (La Niña, cold South Atlantic and Indian Ocean). Again, offline data assimilation was performed to demonstrate the usefulness of the data. The assimilation confirmed and strengthened the circulation anomalies that are already seen in 20CRv3 before assimilating the new data, namely a positive Southern Annular Mode [47]. Much more data will be available for this period and awaits digitization. This should lead to greatly improved reanalyses in future product cycles.

## 8. Conclusions

Historical marine weather data could contribute significantly to better understanding weather and climate processes. Large efforts are currently undertaken, some of which are exemplarily presented in this paper. Studies have also been performed on the improvement in data products such as reanalyses that can be achieved by incorporating some of these data. The importance of data rescue efforts is demonstrated, but much more remains to be done. More 'end-to-end' examples of how specific observations have improved reconstructions of extreme events, or long-term variations and trends, are needed [5,43,47,48].

In the 18th and 19th centuries, and to some extent earlier, vast amounts of information, metadata, and data were collected around the world by state-sponsored commercial and military organizations, proto-scientific institutions and individuals, and by many persons with no more than enthusiastic curiosity to motivate them. Data were often collected for their own sake, and today scientists rely upon that obsession for collecting in order to undertake their research. The process of making that historic data and metadata accessible (data rescue) needs to adopt a similar mindset. For too long, data rescue has been at best a cottage industry, often a component of a larger scientific project. It is time that data rescue was undertaken at a scale 'for its own sake', for only then will the vast amount of untapped historic scientific observations be made available to science. Funding bodies are reluctant to underwrite data collection projects, but this is short-sighted, because science more and more requires data. Funding bodies must be convinced of the value of data rescue and encouraged to properly fund this activity, but it is up to the scientific community at large to bring sufficient pressure to bear on those funding bodies and individuals. Examples of the value (both scientific and financial) of data rescue are needed to help persuade funders. Eventually, the requirement for continuing long-term weather and climate records, where possible and feasible, needs to be highlighted. Tomorrow's long-term records are formed of today's observations. Maintaining long-term observing stations will greatly help future generations to better understand our planet's climate.

**Author Contributions:** Conceptualization, J.L., R.A., C.W., E.H., P.T., A.L., S.B., P.H., K.V. and E.X.; methodology, J.L., R.A., C.W., E.H., P.T., A.L., S.B., P.H., K.V. and E.X.; software, J.L., R.A., C.W., E.H., P.T., A.L., S.B., P.H., K.V. and E.X.; validation, J.L., R.A., C.W., E.H., P.T., A.L., S.B., P.H., K.V. and E.X.; formal analysis, J.L., R.A., C.W., E.H., P.T., A.L., S.B., P.H., K.V. and E.X.; investigation, J.L., R.A., C.W., E.H., P.T., A.L., S.B., P.H., K.V. and E.X.; resources, R.A., C.W., E.H., P.T., A.L., S.B. and E.X; data curation, R.A., C.W., E.H., P.T., A.L., S.B. and E.X; writing—original draft preparation, J.L., R.A., C.W., E.H., P.T., A.L., S.B., P.H., K.V. and E.X; writing—review and editing, J.L., R.A., C.W., E.H., P.T., A.L., S.B., P.H., K.V. and E.X; visualization, R.A., C.W., E.H., A.L., S.B., P.H. and K.V.; supervision, J.L., R.A., C.W., E.H., P.T., A.L., S.B., P.H., K.V. and E.X; project administration, J.L., R.A., C.W., E.H., P.T., A.L., S.B., P.H., K.V. and E.X; funding acquisition, R.A., C.W., E.H., A.L., S.B. and E.X. All authors have read and agreed to the published version of the manuscript.

**Funding:** E.H. and P.T. are funded by the UK NERC GloSAT project and E.H. is supported by the UK National Centre for Atmospheric Science. S.B., E.X., and K.V. were supported by the UK Newton Fund within the framework of the Weather and Climate Service Partnership (WCSSP) South Africa (WCSSP SA22_1.3). A.L. was supported by the NIWA Strategic Science Investment Fund contract CAOA2402 "Fundamental Climate Observations".

**Data Availability Statement:** Historical marine data on convict and settler ships and the Mauritius Project have been sent to the Global Land and Marine Observations Database (GLAMOD) at https://climate.copernicus.eu/global-land-and-marine-observations-database-0 (accessed on 8 January 2024) Information can also be accessed from the International Comprehensive Ocean-Atmosphere Data Set (ICOADS) and the Global Surface Air Temperature (GloSAT) (https://www.glosat.org/ (accessed on 8 January 2024)). Additional information can be accessed from the UK National Archives which holds material over 1000 Royal Navy Medical Officer journals (https://www.nationalarchives.gov.uk/surgeonsatsea/ (accessed on 8 January 2024) and from the UK Ancestry WWW pages https://www.ancestry.co.uk/search/collections/ (accessed on 8 January 2024).

**Conflicts of Interest:** The authors declare no conflicts of interest and the funders had no role in the design of the study; in the collection, analyses, or interpretation of data; in the writing of the manuscript; or in the decision to publish the results.

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
