# Peer review of "The Importance and Scientific Value of Long Weather and Climate Records; Examples of Historical Marine Data Efforts across the Globe"

_climate, doi:10.3390/cli12030039_

Round 1
Reviewer 1 Report
Comments and Suggestions for Authors
This paper highlights the importance of the rescue and preservation of the historical records. I think this paper is worth of publication.
Author Response
We very much thank the positive response of the reviewer, much appreciated. We improved the manuscript by addressing the points raised by reviewers 2 and 3.
Reviewer 2 Report
Comments and Suggestions for Authors
Please see the attached file for a detailed report on the paper

Comments on the Quality of English LanguageAuthor Response
We have addressed the comments/suggestions of reviewer 2 in the attached .pdf file

Reviewer 3 Report
Comments and Suggestions for Authors
This is a position paper where the authors illustrate the need for rescuing historical climatological records gathered in the past, especially marine documents like logbooks. The process of digitization of these documents will make them available to scientific communities. The authors provide a few examples of logbooks and discuss some efforts in this direction, concluding that more funds must be made available for this purpose.
The paper is well-written, and the authors' position is clear and shareable. However, my major concern is how to use these records to reconstruct past climate due to time and spatial discontinuity, which is typical of a naval journey. Doubts may arise about the quality of these records, or at least for some of them. I think that the authors should discuss, at least in one or two sentences in the conclusions, possible drawbacks and how they can be overcome.
Author Response
We have addressed the comments/suggestions of reviewer 3 in the attached .pdf file
